# Nephrotoxic Metal Mixtures and Preadolescent Kidney Function

**DOI:** 10.3390/children8080673

**Published:** 2021-08-02

**Authors:** Yuri Levin-Schwartz, Maria D. Politis, Chris Gennings, Marcela Tamayo-Ortiz, Daniel Flores, Chitra Amarasiriwardena, Ivan Pantic, Mari Cruz Tolentino, Guadalupe Estrada-Gutierrez, Hector Lamadrid-Figueroa, Martha M. Tellez-Rojo, Andrea A. Baccarelli, Robert O. Wright, Alison P. Sanders

**Affiliations:** 1Department of Environmental Medicine and Public Health, Icahn School of Medicine, New York, NY 10029, USA; yuri.levin-schwartz@mssm.edu (Y.L.-S.); maria.politis@mssm.edu (M.D.P.); chris.gennings@mssm.edu (C.G.); daniel.flores@mssm.edu (D.F.); chitra.amarasiriwardena@mssm.edu (C.A.); robert.wright@mssm.edu (R.O.W.); 2Occupational Health Research Unit, Mexican Social Security Institute, Mexico City 06600, Mexico; tamayo.marcela@gmail.com; 3Center for Nutrition and Health Research, National Institute of Public Health, Cuernavaca 62100, Mexico; ivandpantic@gmail.com (I.P.); mmtellez@insp.mx (M.M.T.-R.); 4Department of Developmental Neurobiology, National Institute of Perinatology, Mexico City 06600, Mexico; 5Department of Nutrition, National Institute of Perinatology, Mexico City 06600, Mexico; cruz_tolentino@yahoo.com.mx; 6Department of Immunobiochemistry, National Institute of Perinatology, Mexico City 06600, Mexico; gpestrad@gmail.com; 7Department of Perinatal Health, National Institute of Public Health, Cuernavaca 62100, Mexico; hlamadrid@insp.mx; 8Department of Environmental Health Sciences, Columbia University, New York, NY 10029, USA; ab4303@cumc.columbia.edu; 9Department of Pediatrics, Icahn School of Medicine at Mount Sinai, New York, NY 10029, USA

**Keywords:** arsenic, cadmium, lead, mixture, childhood, kidney

## Abstract

Exposure to metals including lead (Pb), cadmium (Cd), and arsenic (As), may impair kidney function as individual toxicants or in mixtures. However, no single medium is ideal to study multiple metals simultaneously. We hypothesized that multi-media biomarkers (MMBs), integrated indices combining information across biomarkers, are informative of adverse kidney function. Levels of Pb, Cd, and As were quantified in blood and urine in 4–6-year-old Mexican children (n = 300) in the PROGRESS longitudinal cohort study. We estimated the mixture effects of these metals, using weighted quantile sum regression (WQS) applied to urine biomarkers (Umix), blood biomarkers (Bmix), and MMBs, on the cystatin C-based estimated glomerular filtration rate (eGFR) and serum cystatin C assessed at 8–10 years of age, adjusted for covariates. Quartile increases in Umix and the MMB mixture were associated with 2.5% (95%CI: 0.1, 5.0) and 3.0% (95%CI: 0.2, 5.7) increased eGFR and −2.6% (95% CI: −5.1%, −0.1%) and −3.3% (95% CI: −6.5%, −0.1%) decreased cystatin C, respectively. Weights indicate that the strongest contributors to the associations with eGFR and serum cystatin C were Cd and Pb, respectively. MMBs detected mixture effects distinct from associations with individual metals or media-type, highlighting the benefits of incorporating information from multiple exposure media in mixtures analyses.

## 1. Introduction

Chronic kidney disease (CKD) affects between 10 and 15% of the world’s population—a percentage that is rising along with comorbidities such as hypertension, obesity, and diabetes [1]. CKD may have developmental origins from toxic environmental exposures that occur in early childhood [2]. In particular, early-life exposure to common nephrotoxic elements, such as arsenic (As), cadmium (Cd), and lead (Pb), may disrupt nephrogenesis and primary renal developmental processes that are vital for nutrientwaste homeostasis [3]. As, Cd, and Pb are established nephrotoxicants, pervasive in the environment [4,5,6], with substantial evidence of glomerular or tubular toxicity [7,8,9] as well as nephrotoxic effects in children [10,11,12,13,14,15,16]. However, their joint impact on renal development as a mixture is less well known. In this study, we selected As, Cd, and Pb *a priori* based on these reasons and measured levels in blood and urine in a population of healthy children in Mexico City—a population with elevated environmental Pb exposure [17].

Because early-life exposure to nephrotoxicants may lead to kidney function decline, examining exposure to multiple toxicants individually and as a mixture is increasingly important in the context of real-life exposures. For example, blood Pb and Cd exposure have been individually associated with a reduced estimated glomerular filtration rate (eGFR) and co-exposure to both toxicants increases CKD risk [6,18]. Pb and Cd have also been shown to act non-additively, resulting in worse kidney function decline in adults [6,19,20,21]. Additionally, studies in mice show that As and Cd co-exposure exacerbates tubular injury [22]. Despite these consistent results in adults and animal models, the effects of mixed exposures to known nephrotoxicants, such as As, Cd, and Pb, are understudied in the context of early-life exposure and both later childhood as well as adolescent kidney function.

In nearly all human studies, the true internal dose of a chemical, such as a single element, is unknown and must be estimated—most commonly by using exposure biomarker measurements of a compound or its metabolite in a biological medium (e.g., blood or urine) [23]. However, each metal has unique toxicokinetics; i.e., the process by which a chemical enters the body, is metabolized, and excreted. Thus, different metals are distributed differently across different tissues and exposures assessed in a single biological media may not accurately capture exposure to multiple metals simultaneously [24]. This can lead to misleading and uninformative exposure estimates for some chemicals in a mixture when a single exposure media (e.g., blood or urine) is used as a proxy of exposure, thus leading to inaccurate estimates for the effects of the mixture as a whole [25,26]. We have previously reported on a potential method to address this issue, namely, quantifying exposure through multi-media biomarkers (MMBs), estimates of exposure derived using environmental mixture methods across different biological media [27,28,29]. MMBs have proven useful in uncovering the effects of metal mixtures on neurodevelopment [27,28,29]; however, to our knowledge they have not been applied to the role of metal mixtures on kidney function.

In this study, we set out to examine whether early-life exposure to nephrotoxic metal mixtures (NMM) was associated with poorer kidney function, including altered tubular and glomerular parameters. We measured the As, Cd, and Pb levels in the blood and urine from 300 children participating in the Programming Research in Obesity, Growth, Environment, and Social Stressors (PROGRESS) cohort in Mexico City, Mexico. We estimated combined As–Cd–Pb effects, using weighted quantile sum regression (WQS), on renal health, quantified using serum cystatin C and cystatin-C-based eGFR. We compare the mixture effects, estimated using the two media of urine or blood individually, as well as with combined effects when the level of exposure is estimated using MMBs. We propose that MMBs may better highlight the effects of NMM on kidney parameters than any single medium—blood or urine.

## 2. Materials and Methods

### 2.1. Study Participants

The subjects for this study were 300 Mexican children, 8 to 10 years of age, who are participants in the PROGRESS longitudinal birth cohort, which is based in Mexico City. The complete information regarding the enrollment for the parent cohort have been published previously [30]. Briefly, between the years 2007 and 2011, pregnant women who were in their second trimester of pregnancy were recruited for the PROGRESS study through the Mexican Social Security System (Instituto Mexicano del Seguro Social). In total, 948 women delivered a live child into the cohort. Eligibility criteria for the PROGRESS study included the following: being at least 18 years of age, being less than 20 weeks pregnant, being free of both heart and kidney disease, no use of anti-epilepsy drugs or steroids, and no daily consumption of alcohol. The study protocols for PROGRESS were approved by the institutional review boards of the Icahn School of Medicine at Mount Sinai (IRB protocol number: 12–00751), Brigham and Women’s Hospital (IRB protocol numbers: 2006-P-001416 and 2006-P-001792), and the Mexican National Institute of Public Health (IRB protocol number: 560). Written informed consent was obtained from the mothers and at the preadolescent visit children provided assent prior to the collection of samples and all data collection methods were carried out in accordance with the relevant guidelines and regulations. Of the 948 mother–child dyads, 300 had complete information of the exposures, outcomes, and covariates of interest. As of the study visit from which the data used in this study were collected, these children were all healthy and free of kidney or cardiovascular disease, assessed through a maternal questionnaire. Two-sample *t*-tests and chi-squared tests were used to compare the demographics information of this subset with the parent cohort.

### 2.2. Collection of Participant Data

Demographic information was collected from PROGRESS participants. This information includes child sex, child age, self-reported socio-economic status during pregnancy (SES), maternal report of environmental tobacco smoke exposure during pregnancy, and child body mass index (BMI) measured at the same time as the renal outcomes. The maternal report of smoke exposure during pregnancy was collected during the second trimester and indicates whether anyone in the household smoked inside the home. BMI values were categorized into 3 levels, based upon the World Health Organization (WHO) guidelines for children: not overweight or obese (BMI *z*-score < 1), overweight (1 < BMI *z*-score < 2), and obese (BMI *z*-score > 2) [31].

### 2.3. Serum Cystatin C and eGFR

The measurement of serum cystatin C was performed using the Quantikine^®^ human cystatin C immunoassay (R&D Systems, Minneapolis, MN, USA). From the cystatin C measurements, eGFR values were calculated using the following equation: eGFR = 70.69 × (cystatin C)^−0.931^, where cystatin C is in mg/L [32].

### 2.4. Serum Creatinine

Serum creatinine was measured using Creatinine FS reagent and Respons 910 (both by DiaSys, Holzheim, Germany), through the use of the kinetic test without deproteinization according to the Jaffé method [33].

### 2.5. Blood and Urine Metals, and Urine Specific Gravity

The urine and blood samples that were used to measure both the metals and renal biomarkers were collected from the participants between 2012 and 2019. Blood samples were stored at −20 °C and urine samples were stored at −80 °C and shipped to the Laboratory for Environmental Nephrotoxicology at the Icahn School of Medicine at Mount Sinai. For both urine and blood samples, the 200 µL sample was diluted to 10 mL with a diluent solution containing 0.5% HNO_3_, 0.005% Triton X-100, and mixed internal standard in polypropylene trace metal-free Falcon tubes (VWR Metal-Free Centrifuge Tubes). The diluted samples were then analyzed using matrix-matched calibration standards using an Agilent 8900 ICP Triple Quad mass spectrometer (ICP-QQQ) (Agilent Technologies, Inc., Santa Clara, CA, USA) in MS/MS mode with appropriate cell gases to eliminate molecular ion interferences. Internal standards (tellurium for As, rhodium for Cd, and lutetium for Pb) were used to correct for the differences in sample introduction, ionization, and reaction rates in both the plasma and reaction cell. Continuous calibration verification standards, mixed-metal standards at two different concentration levels, procedural blanks, duplicates, and in-house pooled urine or blood samples at three concentration levels were run every 10 samples to monitor the accuracy, recovery rates, and reproducibility of the procedure for each analytic batch. The limits of detection for these metals were between 0.05 ng/mL and 0.2 ng/mL and the limit of quantitation ranged between 0.5 ng/mL and 2 ng/mL. Measurements below the limit of detection (LOD) were replaced with a value of LOD/$\sqrt{2}$. This imputation should have minimal effect on the results because the metal concentrations are grouped into quartiles prior to the statistical analyses. For descriptive statistics we compared the measured blood and urine metal levels to those assessed in a nationally representative United States (US) population during similar years (2009 and 2014) [12].

The measurement of urine specific gravity was performed using a J157HA+ automatic refractometer (Rudolph Research Analytical, Hackettstown, NJ, USA). The urine metal concentrations were corrected for hydration status using the following formula:(1)$$Metal_{Corrected}=Metal_{Original}*\left(\frac{\mu_{SG}-1}{SG-1}\right),$$
where, $Metal_{Corrected}$, is the corrected metal concentration, $Metal_{Original}$, is the original metal concentration, SG, is specific gravity, and, $\mu_{SG}$, is the mean specific gravity value—1.0183 for the subjects in our study [34]. Despite some debate, there is evidence that correction for specific gravity is an improved method to account for hydration status rather than dividing the level of the metal by the concentration of creatinine [35].

### 2.6. Statistical Analyses

#### 2.6.1. Weighted Quantile Sum Regression

WQS is a statistical method to assess the joint effect of multiple predictors, grouped into quantiles, on an outcome. In this study, we applied WQS in two ways (1) to assess the mixture effect of As, Cd, and Pb and (2) to assess the contribution of each biomarker to the MMB. If we consider M predictors, grouped into quantiles, $q_{m}$, and our outcome, y, the general WQS model is expressed as
(2)$$g\left(\mu \right)\approx \beta_{0}+\beta_{1}\left(\overset{M}{\underset{m=1}{\sum }}w_{m}q_{m}\right)+z^{T}\phi ,$$
where g(μ) is the link function, and for this study the identity function, $w_{m}$, is the weight for the mth predictor, and z is the set of covariates with regression coefficients, ϕ. The weights are the mean weight across 1000 bootstrapped datasets and constrained to be both non-negative and sum to one. These constraints mean that the weights represent the relative contribution of each predictor to the overall association; higher weights indicate a stronger contribution. We validate the assumptions of linearity and directional homogeneity in WQS through (i) visual inspection of residuals and (ii) comparison of the fit of the linear models to nonlinear (quadratic) models. For improved consistency of our results, we did not separate our dataset into training and testing datasets. A more detailed discussion of WQS has been described previously [36].

#### 2.6.2. Multi-Media Biomarkers

MMBs combine exposure information across multiple biomarkers into constructs that have been proposed to better reflect body burden [27,28]. To derive the MMBs in this study, we integrated the metal levels from blood and urine using WQS for each metal. This derivation enables the determination of weights that highlight the contribution of each biomarker to the MMB. When using MMBs to measure the effects of metals on renal health, four WQS models were used, one for each metal—As, Cd, and Pb—to derive the MMBs and one to estimate the mixture effect of the MMBs. Therefore, these analyses are hierarchical WQS analyses; the first level of this analysis is a WQS analysis across media for a single metal and the second level is a WQS analysis across metals [28,37].

#### 2.6.3. Data Analysis Workflow

After converting the exposure levels to quartiles, we conducted four (4) main types of analysis for each outcome, serum cystatin C and cystatin C-based eGFR. (I) In our first set of analyses, we performed linear regressions where the outcome was the measure of kidney function and the exposure was the metal level in each biomarker. We repeated these analyses for each combination of metal (As, Cd, and Pb) and medium (blood and urine), resulting in a total of six regression models. (II) In the second set of analyses, we used WQS to estimate the combined effect of As, Cd, and Pb on renal parameters, where the levels of each metal were quantified in each medium. Thus, we estimated separate WQS indices for each medium individually (i.e., a WQS index for the metals in blood: B_mix_; and a WQS index for the metals in urine: U_mix_). Then, we assessed the associations between these two indices and the outcomes using linear regressions. (III) In the third set of analyses, we estimated the MMBs as described in the previous section, categorized them into quartiles, applied a second WQS on the MMBs, and assessed the association between this second level index and the outcome using linear regressions. (IV) In our final set of analysis, we combined all combinations of the exposure biomarkers and metals using a single, omnibus WQS index and measured the association between this index and the outcome using a linear regression [29]. All regression coefficients were exponentiated to represent percent—fold—changes in the outcome for a 1-quartile increase in the corresponding exposure level. All models were adjusted for child age (years), child sex, indoor tobacco smoke exposure (yes vs. no), SES (lower vs. medium vs. higher), and BMI (normal: BMI *z*-score < 1 vs. overweight: 1 < BMI *z*-score < 2 vs. with obesity: BMI *z*-score > 2). The reference categories for SES and BMI were lower and normal (1 < BMI *z*-score < 2), respectively. We selected these covariates *a priori* based on a review of the factors affecting both kidney function and metal levels. In our analyses, we grouped the predictors into quartiles. We constrained the directionality of our associations in both the positive and negative directions and present the results from the models with significant associations. All analyses were performed using SAS 9.4.

## 3. Results

### 3.1. Characteristics of the Study Participants

The demographic information of the 300 participants in this study is shown in Table 1. More than half of the population were boys and the average age was 9.6 years. Roughly half of the participants, 54% (n = 164), had a normal BMI, 24% (n = 71) were overweight, and 22% (n = 67) with obesity [31]. Descriptive statistics for the metal levels are shown in Table 2. Notably, the mean levels of blood Cd and urine Cd were significantly lower than compared with a representative sample of children from the US in similar years [12]. Additionally, the mean levels of blood Pb, urine As, and urine Pb were significantly higher for the children in this study than a representative sample of US children [12]. The demographic information of the subset of subjects used in this study did not deviate significantly from the parent cohort (data not shown).

### 3.2. Individual Metals Exposure Is Associated with Kidney Biomarkers

As shown in Figure 1 and Figure 2, urinary Cd was associated with eGFR and cystatin C. A one quartile increase in urinary Cd was associated with an increase in eGFR of 2.3% (95% CI: 0.1%, 4.5%) and a decrease in cystatin C of 2.4% (95% CI: 0.2%, 4.7%). No other metals were individually associated with either eGFR or cystatin C.

### 3.3. Metal Mixtures Are Associated with Kidney Biomarkers

The mixtures of metals in urine, MMBs, and the omnibus combination of all metals and biomarkers were all associated with eGFR (Figure 3). Quartile increases in U_mix_, the mixture of MMBs, and the omnibus combination of all metals and biomarkers were associated with increases in eGFR of 2.5% (95% CI: 0.1%, 5.0%), 3.0% (95% CI: 0.2%, 5.7%), and 3.2% (95% CI: 0.1%, 6.4%), respectively. From Table 3, Cd was the strongest contributor to these associations, with a corresponding weight of 84% in U_mix_, 74% in the MMB mixture, and 73% in the omnibus mixture. Both MMBs and the omnibus mixture enable further exploration of the relative importance of urine and blood to the observed associations. As shown in Table 4, urine metals contributed the majority of the weight to the observed associations, with 66% of the MMB mixture association and 67% of the omnibus mixture association attributed to urine metals.

Correspondingly, U_mix_, the mixture of MMBs, and the omnibus combination of all metals and biomarkers were associated with decreased serum cystatin C (Figure 4). A one quartile increase in U_mix_, the MMB index, and the omnibus index were associated with a corresponding 2.6% (95% CI: 0.1%, 5.1%), 3.1% (95% CI: 0.2%, 5.8%), and 3.3% (95% CI: 0.1%, 6.5%) decrease in serum cystatin C, respectively. As was the case with the eGFR results, Cd was the strongest contributor to the association (Table 5), with this single metal contributing 84% of the U_mix_ weights, 73% of the MMB mixture weights, and 74% of the omnibus mixture weights. Urine was also the strongest contributing medium to the MMB and omnibus indices, with 68% and 67% of the weights (Table 6).

### 3.4. Metals and Serum Creatinine

We also investigated the effects of metals both individually and as part of mixtures on serum creatinine after adjusting for the same set of covariates as in the main analyses (Appendix A). The results are shown in the supplementary materials. When assessing the effects of the metals individually, only urine Pb was associated with serum creatinine. A quartile increase in urine Pb was associated with a 0.01 mg/dL (95% CI: 0.00, 0.02).

When assessing the effects of metal mixtures on serum creatinine, B_mix_, U_mix_, the mixture of MMBs, and the omnibus mixture were all associated with decreased serum creatinine (Appendix A). Quartile increases in B_mix_, U_mix_, the mixture of MMBs, and the omnibus mixture were associated with decreases in serum creatinine of 0.02 mg/dL (95% CI: 0.00, 0.03), 0.01 mg/dL (95% CI: 0.00, 0.02), 0.02 mg/dL (95% CI: 0.01, 0.04), and 0.02 mg/dL (95% CI: 0.01, 0.04), respectively. From Appendix A, the metal contributing the most weight to these associations was Pb, with 44%, 88%, 48%, and 54% of the weight of the associations for B_mix_, U_mix_, the mixture of MMBs, and the omnibus mixture, respectively. Overall, blood was the biomarker that contributed most to the MMB and omnibus mixture effects (Appendix A). Approximately 67% of the MMB mixture effect and 62% of the omnibus mixture effect were from blood metals.

## 4. Discussion

In this study, we examined the mixture effect of As, Cd, and Pb on markers of kidney function using WQS and MMBs. Overall, we found that the mixture effects differed depending on the medium used to measure exposure and that the MMB approach as well as the omnibus mixture consistently uncovered stronger mixture effects than those found using the biomarkers individually. This highlights the advantages of incorporating exposure information from multiple biomarkers, reducing errors in the estimation of exposure. In this work, we report modest increases in the metal mixture associated eGFR (~3% increases), largely driven by Cd. While these findings are not clinically of concern, early increases in filtration leading to hyperfiltration are beginning stages of eventual eGFR decline in some CKD [38,39]. Therefore, long-term follow-up among at risk groups with early-life elevated eGFR may be prudent.

In models for U_mix_, the mixture of MMBs, and the omnibus mixture, urine Cd was consistently identified as the strongest contributor (73–84%) to the relationship with a higher subsequent eGFR. A prior study of As, Cd, Pb, and mercury (Hg) mixtures among US adolescents in NHANES similarly identified a positive association between a similar mixture of urinary metals and serum creatinine-derived eGFR wherein the top weighted metals contributing to the association were Hg (61%), Cd (17%), and As (13%) [12]. While we did not assess Hg in this study, the relative contributions for the metals that overlap across the two studies, namely As, Cd, and Pb, is identical across studies. Although generally in alignment with our results, differences between the reported findings could be due to study design, source population exposure sources and characteristics, younger age of participants in the PROGRESS cohort, and differences in available metals and kidney function measures between the two studies. A study of metal mixtures assessed in blood from a large sample of US adults reported significant associations with multiple measures of worse kidney function, including reduced eGFR [40]. The strongest contributors to those associations included blood Pb and Cd. In our study, we observed no significant associations with B_mix_, similar to findings in the prior study of adolescents in the US [12]. The differences between the reported findings for blood metal mixtures may be due to the age difference between the subjects or the differences in the mixture methods used. The present study, and the one reporting no associations for the blood metals and renal health, studied children and adolescents, and both used WQS [12], while there were associations observed for adults detected using Bayesian kernel machine regression (BKMR) [40]. The inconsistencies of the metal mixture effects on eGFR depending on the exposure biomarker has been reported previously [41].

While there have been relatively few papers exploring the impact of metal mixtures on kidney function, especially in adolescents, there is a larger literature on these metals individually and kidney disease or dysfunction. For example, in two cross-sectional studies of adults and adolescents, increased levels of urine Cd were associated with increased eGFR [42,43]. Additionally, in a cross-sectional study of 1253 adolescents/young adults in the US, higher total urinary As was associated with higher eGFR [44]. Among studies that examined individual metal levels in blood and kidney health, two reported associations between blood Pb increased eGFR [45,46]. A potential mechanism to explain these results is that increased exposure to metals leads to subclinical glomerular damage, resulting in overcompensation observed as increased filtration (hyperfiltration) [41,43,44,47,48]. The cross-sectional design of the vast majority of prior studies makes temporal inference difficult. For example, chronic exposure to one or more toxicant metals can lead to decreased kidney filtration capacity due to reduced capacity for toxicant excretion, thus resulting in renal accumulation of multiple toxicants and further explaining mixture effects [12]. In this work, we considered a relatively simple mixture of three established nephrotoxic metals selected a priori; however, issues of bias in exposure estimation become increasingly strong as the number of elements in the mixture increases. The reason for this is that a single biomarker is less likely to be an accurate measure of exposure for every element in a mixture with a large number of elements than in a simple mixture.

Our study has multiple strengths. The participants in the PROGRESS cohort are free of clinical renal disease. We measured metal exposure when the participants were 4–6 years of age and kidney outcomes when the participants were 8–10 years of age, thus enabling us to examine temporal relationships between exposure and outcome. Our use of multiple exposure biomarkers, uncommon in epidemiological studies, enabled us to explore how the association between metal mixtures and kidney function parameters differs depending on the choice of exposure media. Using MMBs allows us to minimize the potential bias that can be introduced through the selection of a single biological media to measure the mixture effects of As, Cd, and Pb. We used four complementary modeling techniques to explore associations between metals and renal parameters: (i) individual metal regressions, (ii) media-specific mixture analyses (U_mix_ and B_mix_); (iii) MMB mixture analyses, and (iv) an omnibus mixture of metals and biomarkers. We found strong consistency across these different modeling methods, increasing our confidence in our results. We used an established mixtures method, WQS, which assesses the joint effect of multiple metals on kidney function while being robust to exposure collinearly. Additionally, our use of WQS facilitates the estimation of weights highlighting the relative contributions of different metals to the mixture effect as well as the relative contributions of different exposure biomarkers to detect these associations. Although Cd dominated the weighted indices, there was a contribution from other metals, supporting the concept of a nephrotoxic mixture effect.

Our study also has some limitations. We assessed only eGFR at a single time point; follow-up studies in mid-adolescence will provide additional information about the eGFR trajectory. The PROGRESS cohort did not collect information about the hour of the day of urine collection as well as whether urine samples analyzed for metals were the first urine of the day, and so we were unable to account for this in our analyses. However, the samples were collected in the morning of each visit and, thus, constitute either the first or second urine of the day. Subsequent PROGRESS study visits are collecting information on time of urine collection, enabling its incorporation into analyses moving forward. We used spot urine samples for our analyses and do not have information about 24-h creatinine clearance. We were not able to measure iohexol clearance, considered the “gold standard” of measured GFR, as this may not be practical in large population-based studies. In our WQS analyses, we did not split our data into separate training and testing datasets; therefore, our results could suffer from a lack of generalization. As with all observational epidemiological studies, replication in other populations is needed to validate our results. Finally, as with all studies of metals and renal health, reverse causality is a major concern. As kidney function declines, the changes in metal excretion rates can lead to measured increases in urinary metal levels [49]. However, the probability of this may be minimal in this study as the participants do not have any clinically-diagnosed kidney disease or dysfunction.

## 5. Conclusions

There is a need to understand how early-life environmental determinants of kidney dysfunction may lead to diseases such as CKD or hypertension. Mitigating exposures in early-life that contribute to impaired kidney function is critical to develop cost-effective public health efforts to protect population health prior to the onset of chronic disease. This study provides insight into the association between nephrotoxicant exposure and kidney function, as well as providing an innovative and powerful approach to estimate exposure based on multi-media biomarkers (MMBs). We found that the mixture effects were different based upon the exposure biomarker used and that the MMB approach as well as the omnibus mixture detected stronger associations than those found using the biomarkers individually. Understanding the effects of joint exposure to nephrotoxicants on renal health will facilitate comprehension of the mechanisms involved in metal exposure and filtration in the kidney.

## Figures and Tables

**Figure 1 children-08-00673-f001:**
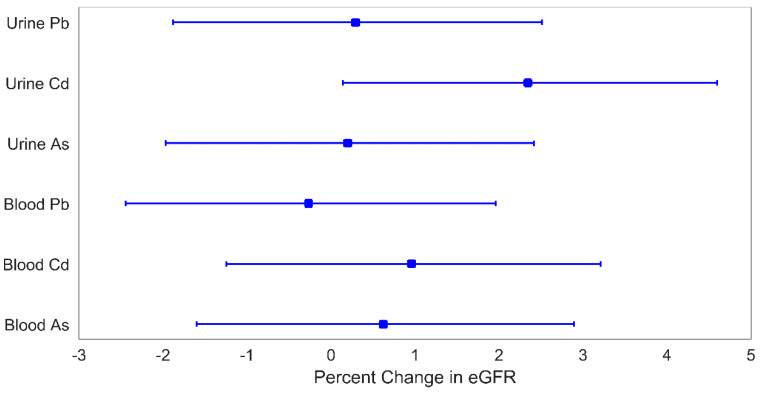
Percent change in eGFR for a 1-quartile increase in metal levels. Plotted points are beta coefficients and lines are 95% confidence intervals. All models were adjusted for child age (years), child sex, indoor tobacco smoke exposure (referent group: no), SES (referent group: lower), and BMI (referent group: BMI *z*-score < 1). The *p*-values for these associations are: 0.79 (urine Pb), 0.04 (urine Cd), 0.86 (urine As), 0.81 (blood Pb), 0.40 (blood Cd), and 0.59 (blood As).

**Figure 2 children-08-00673-f002:**
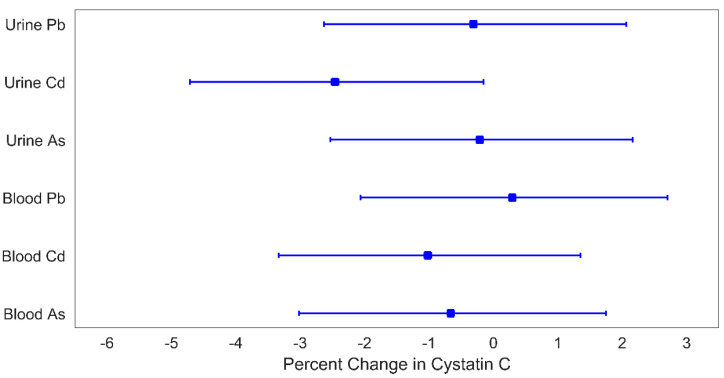
Percent change in cystatin C for a 1-quartile increase in metal levels. Plotted points are beta coefficients and lines are 95% confidence intervals. All models were adjusted for child age (years), child sex, indoor tobacco smoke exposure (referent group: no), SES (referent group: lower), and BMI (referent group: BMI *z*-score < 1). The *p*-values for these associations are: 0.79 (urine Pb), 0.04 (urine Cd), 0.86 (urine As), 0.81 (blood Pb), 0.40 (blood Cd), and 0.59 (blood As).

**Figure 3 children-08-00673-f003:**
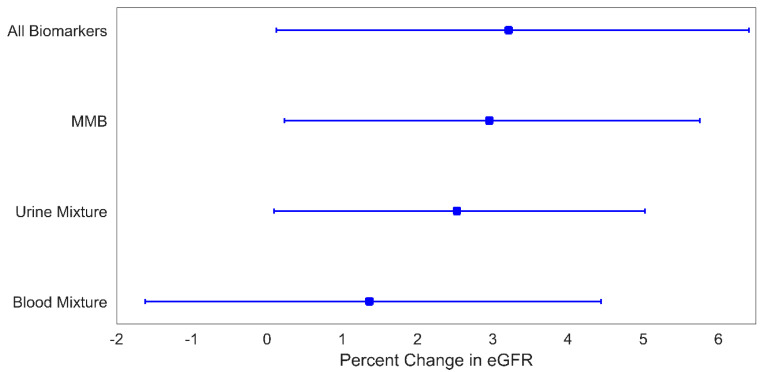
Percent change in eGFR for a 1-quartile increase in metal mixture levels. Plotted points are beta coefficients and lines are 95% confidence intervals. All models were adjusted for child age (years), child sex, indoor tobacco smoke exposure (referent group: no), SES (referent group: lower), and BMI (referent group: BMI *z*-score < 1). The *p*-values for these associations are: 0.04 (all biomarkers), 0.03 (MMB), 0.04 (urine mixture), and 0.37 (blood mixture).

**Figure 4 children-08-00673-f004:**
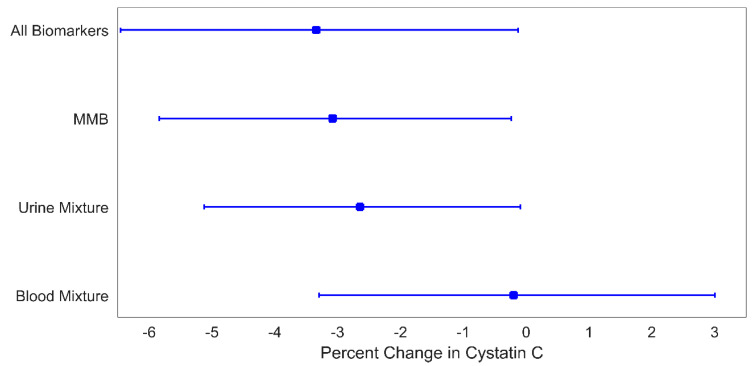
Percent change in cystatin C for a 1-quartile increase in metal mixtures levels. Plotted points are beta coefficients and lines are 95% confidence intervals. All models were adjusted for child age (years), child sex, indoor tobacco smoke exposure (referent group: no), SES (referent group: lower), and BMI (referent group: BMI *z*-score < 1). The *p*-values for these associations are: 0.04 (all biomarkers), 0.03 (MMB), 0.04 (urine mixture), and 0.90 (blood mixture).

**Table 1 children-08-00673-t001:** Demographic information and descriptive statistics of the PROGRESS subjects in this study.

Demographics	Category	n (%)
Total		300 (100%)
Child Sex	Male	158 (53%)
	Female	142 (47%)
Indoor Tobacco Smoke Exposure during Pregnancy	No	207 (69%)
	Yes	93 (31%)
Socio-economic Status during Pregnancy	Lower	161 (54%
	Medium	112 (37%)
	Higher	27 (9%)
Child Body Mass Index	Normal	162 (54%)
	Overweight	71 (24%)
	Obese	67 (22%)
		Mean ± SD
Child Age (years)		9.6 ± 0.6
Serum Creatinine (mg/dL)		89.9 ± 35.6
Estimated Glomerular Filtration Rate (Cystatin C, mL/min/1.73 m^2^)		100.5 ± 23.1
Serum Cystatin C (ng/mL)		724 ± 172

**Table 2 children-08-00673-t002:** Distribution of blood and urine metal concentrations in this sub-cohort, measured at 4–6 years of age (n = 300).

Metal	25th Percentile	Median	75th Percentile	Mean ± SD	Mean ± SD [12] ^a^
Blood As (ng/dL)	36.9	45.6	62.2	64.5 ± 85.6	N/A
Blood Cd (ng/dL)	5.4	7.2	9.5	7.8 ± 3.6	20 ± 46
Blood Pb (μg/dL)	1.3	1.6	2.4	2.3 ± 2.4	0.7 ± 1.0
Urine As (μg/L)	8.9	13.6	20.1	19.0 ± 25.8	13.6 ± 62.0
Urine Cd (ng/L)	44.9	63.5	92.3	75.2 ± 67.8	90 ± 130
Urine Pb (μg/L)	0.9	1.5	2.5	2.3 ± 4.2	0.4 ± 0.5

^a^ Study of all NHANES participants between the ages of 12 and 19 years between 2009 and 2014, used for comparison.

**Table 3 children-08-00673-t003:** Weights highlighting the contribution of each metal to the mixture effect of metals on eGFR. Larger weights indicated higher contributions to the mixture effect.

	Blood Mixture (B_mix_)	Urine Mixture (U_mix_)	MMB	All Biomarkers
As	26%	8%	22%	19%
Cd	10%	84%	74%	73%
Pb	65%	8%	4%	8%

**Table 4 children-08-00673-t004:** Weights highlighting the contribution of each biomarker to the mixture effect of metals on eGFR. Larger weights indicated higher contributions to the mixture effect.

	As MMB	Cd MMB	Pb MMB	MMB Overall	All Biomarkers
Blood	69%	22%	68%	34%	33%
Urine	31%	78%	33%	66%	67%

**Table 5 children-08-00673-t005:** Weights highlighting the contribution of each metal to the mixture effect of metals on Cystatin C. Larger weights indicated higher contributions to the mixture effect.

	Blood Mixture (B_mix_)	Urine Mixture (U_mix_)	MMB	All Biomarkers
As	23%	8%	22%	19%
Cd	15%	84%	73%	73%
Pb	62%	8%	5%	8%

**Table 6 children-08-00673-t006:** Weights highlighting the contribution of each biomarker to the mixture effect of metals on cystatin C. Larger weights indicated higher contributions to the mixture effect.

	As MMB	Cd MMB	Pb MMB	MMB Overall	All Biomarkers
Blood	70%	22%	26%	32%	33%
Urine	30%	78%	74%	68%	67%

## Data Availability

The data that was used in this study can be made accessible to researchers upon appropriate request with the following restrictions to ensure the privacy of human subjects. Note that access to the data is limited due to a data sharing agreement approved by the IRB at Mount Sinai. Researchers that are interested in accessing PROGRESS data must send their resume/CV as well as CITI training certificates to the IRB chair, Ilene Wilets (ilene.wilets@mssm.edu). They must also submit a data analysis plan to the Principal Investigators for PROGRESS; Robert O. Wright (robert.wright@mssm.edu), Martha Tellez-Rojo (mmtellez@insp.mx), and Andrea Baccarelli (andrea.baccarelli@columbia.edu). Once this process is completed, the PROGRESS data analyst, Nia McRae (nia.x.mcrae@mssm.edu) will send a de-identified dataset via Box, a secure data sharing platform.

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
