# Peer review of "Nephrotoxic Metal Mixtures and Preadolescent Kidney Function"

_children, 2021, doi:10.3390/children8080673_

Round 1

Reviewer 1 Report

In this research paper, the authors aimed to investigate the link between impaired kidney function and metal exposure in pre-adolescent subjects. They used an interesting approach to estimating exposure based on multimedia biomarkers. 
The study is well designed and developed, the statistical analysis is adequately performed. 
The authors also properly cite and mention data from previous studies on the field. 

But minor corrections should be performed: 

  • References section – spaces, fonts, and instructions must be respected.
  • English style should be improved in style and grammar.
  • A list of abbreviations should be included. 

Author Response

Reviewer 1:

In this research paper, the authors aimed to investigate the link between impaired kidney function and metal exposure in pre-adolescent subjects. They used an interesting approach to estimating exposure based on multimedia biomarkers.

The study is well designed and developed, the statistical analysis is adequately performed.

The authors also properly cite and mention data from previous studies on the field.

We thank the reviewer for their time as well as their comments.

But minor corrections should be performed:

References section – spaces, fonts, and instructions must be respected.

We have updated the references to be in the appropriate format for the journal.

English style should be improved in style and grammar.

We have improved the style and grammar

A list of abbreviations should be included.

We now include a list of abbreviations.

Reviewer 2 Report

The work submitted by Levin-Schwartz et al. tries to evaluate how exposure to different metals affects kidney function in the pediatric population. In my opinion, this is a very interesting article for the field of Environmental and Clinical Toxicology with sufficient quality to be published. The introduction presented and the methodology designed is outstanding (especially the statistical section). However, I think there are some important aspects that need to be corrected or improved:
Major points:
- In the Methodology section, they talk about how the BUN is quantified ... why is it not presented in the Results? this biomarker is never discussed again ... the same analysis that has been done with other renal markers should be performed to observe the influence of different metals and parameters on BUN (as is done with eGFR, cystatin and creatinine).
- In the same way, in Results, plasma creatinine is presented ... why is it not explained in Methodology how it was quantified?
Minor points:
- The acronym BUN is not defined in the text.
- The Methodology section indicates that the samples were stored at 4ºC, but ... how were they collected? What anticoagulant was used? How long were they kept at 4ºC? since the components of that sample can degrade at that temperature if they are kept for a long time without freezing.
- Lines 125-126: the sentence is badly written.
- Line 130: the subscript must be written in the chemical formula.
- Table 1: N should be n (it is more correct to indicate the sample size).
- Table 1: I think the acronyms should be defined in the table heading.
- Table 2: Include the name of reference 12, not just the number (it is very strange to see "from 12").

Author Response

The work submitted by Levin-Schwartz et al. tries to evaluate how exposure to different metals affects kidney function in the pediatric population. In my opinion, this is a very interesting article for the field of Environmental and Clinical Toxicology with sufficient quality to be published. The introduction presented and the methodology designed is outstanding (especially the statistical section). However, I think there are some important aspects that need to be corrected or improved:

We thank the reviewer for their time as well as their comments.

Major points:

- In the Methodology section, they talk about how the BUN is quantified ... why is it not presented in the Results? this biomarker is never discussed again ... the same analysis that has been done with other renal markers should be performed to observe the influence of different metals and parameters on BUN (as is done with eGFR, cystatin and creatinine).

We apologize for the confusion. We did not find any associations between the metals individually or as part of a mixture and BUN. We do not feel that presenting these findings does not add meaningfully to the conclusions of the manuscript. For this reason, we have removed the text related to BUN from the methodology Section.

- In the same way, in Results, plasma creatinine is presented ... why is it not explained in Methodology how it was quantified?

We have now added text to the methodology section describing the quantification procedure for plasma creatinine. 

Minor points:

- The acronym BUN is not defined in the text.

We apologize for the confusion. We have removed the section about BUN from the manuscript.

- The Methodology section indicates that the samples were stored at 4ºC, but ... how were they collected? What anticoagulant was used? How long were they kept at 4ºC? since the components of that sample can degrade at that temperature if they are kept for a long time without freezing.

We apologize for the confusion. This was a typo. The blood samples were stored at -20 oC and not at +4 oC. We have corrected this issue in the methodology section.

- Lines 125-126: the sentence is badly written.

We have updated this sentence to make it clearer.

- Line 130: the subscript must be written in the chemical formula.

We now include the subscript in the chemical formula.

- Table 1: N should be n (it is more correct to indicate the sample size).

We have changed n/N to m/M to avoid any confusion between the number of predictors (m/M) and the number of samples (n/N). We have updated this throughout the methodology section.

- Table 1: I think the acronyms should be defined in the table heading.

We have added definitions of the acronyms in Table 1.

- Table 2: Include the name of reference 12, not just the number (it is very strange to see "from 12").

We have now included the name of the reference in Table 2. 

Round 2

Reviewer 2 Report

Changes have been made successfully